# Risk of Benign and Malignant Thyroid Disorders in Subjects Treated for Paediatric/Adolescent Neoplasia: Role of Morphological and Functional Screening

**DOI:** 10.3390/children8090767

**Published:** 2021-08-31

**Authors:** Giulia Sapuppo, Martina Tavarelli, Emanuela Cannata, Milena La Spina, Marco Russo, Claudia Scollo, Angela Spadaro, Romilda Masucci, Luca Lo Nigro, Giovanna Russo, Andrea Di Cataldo, Gabriella Pellegriti

**Affiliations:** 1Endocrinology, Garibaldi-Nesima Medical Center, Department of Clinical and Experimental Medicine, University of Catania, 95122 Catania, Italy; giuliasapuppo@hotmail.it; 2Endocrinology, Garibaldi-Nesima Medical Center, Via Palermo 636, 95122 Catania, Italy; martinatava@hotmail.it (M.T.); mruss@hotmail.it (M.R.); claudiascollo@hotmail.com (C.S.); spadaro_angela@virgilio.it (A.S.); 3Onco-Ematology Unit, Department of Clinical and Experimental Medicine, University of Catania, 95122 Catania, Italy; e.cannata80@gmail.com (E.C.); mlaspina@unict.it (M.L.S.); lucalonigro1968@gmail.com (L.L.N.); diberuss@unict.it (G.R.); Adicata@unict.it (A.D.C.); 4Garibaldi-Nesima Medical Center, Department of Surgical Oncology, 95122 Catania, Italy; romilda.masucci@alice.it

**Keywords:** childhood/adolescent cancer, thyroid disease, thyroid cancer, secondary thyroid cancer, chemotherapy and radiotherapy

## Abstract

Background: Patients treated for paediatric/adolescent (P/A) neoplasia have a high incidence of both benign and malignant thyroid diseases. Given the high incidence of sequelae, literature data show a clinical benefit of morpho-functional thyroid screening in paediatric/adolescent cancer survivors and a careful lifetime follow-up. Patients and methods: The incidence of thyroid alterations was evaluated in a consecutive series of 343 patients treated with chemotherapy (CHE) and radiotherapy (RTE) or only with CHE for P/A tumours between 1976 and 2018 (mean age at time of primary paediatric malignancy 7.8 ± 4.7 years). All patients underwent thyroidal morpho-functional evaluation between 2000 and 2019. Results: 178 patients (51.9%) were treated only with CHE and 165 (48.1%) with CHE+RTE. A functional and/or structural thyroid disease was diagnosed in 147 (42.5%; 24.2% in CHE and 62.4% in CHE+RTE group; *p* = 0.0001). Of note, 71 (20.7%) patients with no evidence of disease at first evaluation developed a thyroid alteration during the follow-up. Primitive hypothyroidism was diagnosed in 54 patients (15.7%; 11.2% in CHE vs. 20.6% in CHE+RTE group; *p* = 0.01) and hyperthyroidism in 4. Sixty-three patients developed thyroid nodules (18.4%; 4.0% in CHE and 14.1% in CHE+RTE group; *p* < 0.001); thyroid cancer was diagnosed in 30 patients (8.7%; 4.5% in CHE and 12.4% in CHE + RTE group; *p* = 0.007). Conclusions: In patients treated with CHE+RTE, the prevalence of hypothyroidism and nodular pathology, both malignant and benign, were significantly greater than in patients treated with CHE. However, also in the CHE group, the frequency of thyroid disease is not negligible and the pathogenetic mechanisms remain to be clarified. Our data suggest the clinical benefit of morpho-functional thyroid screening in P/A cancer survivors.

## 1. Introduction

Over the last decades, advances in the medical treatment of paediatric malignancies have led to improved outcomes with a significant reduction in child mortality and a current five year survival rate exceeding 80% [1]. The increased number of childhood cancer survivors in the last few decades has drawn an increased attention to the identification and treatments of long-term sequelae, as second cancers, cardiovascular disease, lung fibrosis, renal dysfunction, severe musculoskeletal disease, neurocognitive defects and endocrine and reproductive dysfunctions. The risk of late and chronic health sequelae, which may appear early in childhood or adolescence or many years later during adulthood, is high and it is estimated that 60–75% of survivors of childhood cancer will develop at least one late effect as a direct result of their treatment [2]. The endocrine sequelae of childhood cancer survivors are among the most common side effects that are observed in 40–60% of treated patients [3,4]. They are a consequence of exposure to therapies such as chemotherapy (CHE) and radiotherapy (RTE), alone or in combination, and include hypothalamus-pituitary axis, thyroid gland, gonadal and pancreas alterations [3,4,5,6].

The thyroid gland is highly sensitive to the effects of ionizing radiation as confirmed by the high incidence of thyroid disease in Japanese atomic bomb survivors [7,8], in subjects exposed to the nuclear fallout in Chernobyl [9] and more recently in the Fukushima nuclear accident [10].

Several risk factors have been associated with radiation-induced thyroid dysfunction. These risks include young age at the time of irradiation, female gender, radiation dose, genetic predisposition, fractionated dose (hyperfractionated RTE) and the irradiation field [3,9].

Recently, an increased incidence of hypothyroidism and thyroid cancer has also been reported in patients receiving only CHE with alkylating agents (busulfan, cyclophosphamide), which can cause DNA damage with detrimental effects on cell biology [11,12]. It is still unclear whether CHE+RTE has an additional negative effect on the thyroid [13,14,15].

To evaluate this issue, as well as the frequency and type of thyroid disorders, we studied patients treated with CHE+RTE and CHE alone for childhood/adolescence malignancies. Since 2000, a screening program was established at the Thyroid Clinic of the Garibaldi-Nesima Medical Centre in Catania, Italy. All patients “off-therapy” were previously treated and followed up at the Paediatric Haematology and Oncology Unit at the University of Catania, Italy.

## 2. Patients and Methods

Starting in January 2000, 343 patients (167 females (F) and 176 males (M)) previously treated with CHE alone and CHE+RTE in the years 1976–2018 for paediatric cancer at the Paediatric Haematology and Oncology Unit of the University of Catania, Italy, attended a periodic screening program for thyroid disease at the Thyroid Clinic at the Endocrinology Division, Garibaldi-Nesima Medical Centre of Catania.

The types of childhood cancer are shown in Table 1.

Mean age at the time of detection of primary malignancy was 7.8 ± 4.7 years (median 7.2 years), and the mean time interval from the time of diagnosis to the first control visit for thyroid diseases was 8.2 ± 4 years (median 6.1 years) while the time interval to the last control visit was 13.8 ± 9 years (median 10.3 years).

According to the Italian Association of Pediatric Hematology and Oncology (AIEOP) protocols, 178 patients (51.9%) were treated with CHE according to tumour diagnosis and stage, and 165 patients (48.1%) were treated with CHE+RTE. The mean radiation dose received was 29.2 ± 17 Gy, and the mean age at the time of irradiation was 9.9 ± 4.7 years. The site, dose and time of irradiation changed according to the type and location of the tumour. To reduce the risk of side effects like bone marrow aplasia, particularly in patients who were given concomitant CHE, a hyperfractionated scheme of treatment five days a week was used in most cases. RTE was administered to the brain and the spinal cord for acute lymphocytic leukaemia (neurological prophylaxis) and to the neck, mediastinum, chest and abdomen for lymphomas (in relation to the stage).

At the first visit, anamnestic data were collected, and all patients underwent clinical and thyroid ultrasound examination; TSH, FT3, FT4, calcitonin, anti-thyroid peroxidase (AbTPO) and anti-thyroglobulin (AbTg) antibodies were measured. TRAb (anti-TSH receptor antibodies) were also measured in patients with hyperthyroidism.

When a morphological and/or functional thyroid alteration (hypothyroidism and hyperthyroidism, thyroid nodules and thyroid carcinoma) was found, patients were followed according to national and international guidelines. Patients without thyroid abnormalities were followed yearly.

Thyroid ultrasound examination included the evaluation of ultrasound patterns of the thyroid gland, subdivided into grade 1, 2 or 3 (normal, moderately hypoechoic or diffusely hypoechoic, respectively). Nodules were evaluated by EU-TIRADS classification [16], and when greater than 1.0 cm in size or smaller but with ultrasound suspicious characteristics were examined by fine-needle aspiration biopsy (FNAB). In about half of the patients with nodules, FNAB was not performed as nodules were smaller than 1 cm (32 of 63 patients) and in the remaining patients for individual decision. Tumors were staged according to the pTNM 7th edition [17]: T (the extent of the primary tumor) and N (regional LN metastases) were assessed on the basis of the pathological examination and M (distant metastases) according to the first post-surgical ^131^I -whole body scintigraphy (WBS) and/or other imaging modalities. ^131^I treatment activity was empirically 1 mCi/Kg of body weight and was administered following levothyroxine withdrawal in 23 patients.

All procedures involving human participants were in accordance with the ethical standards of the institutional research committee and with the Helsinki declaration as revised in 2013. Ethical review and approval were waived for this retrospective study. Patients or legal tutors of all the patients signed a written consent form for the use of their data in clinical research studies.

### Statistical Analysis

Categorical variables were expressed as frequencies and percentages and analyzed using the chi-square test with Yates’ correction or Fisher’s test for small samples. Normally distributed quantitative variables were expressed as mean ± standard deviation (SD), while non-normally distributed variables were expressed as median and interquartile range (IQR). Quantitative variables were analyzed by the Student’s t-test or the Mann-Whitney U test. Data analysis was performed using Stata version 13.1. A *p* value <0.05 was considered statistically significant for all analyses.

## 3. Results

Patients were divided into two groups: a CHE group and a CHE+RTE group.

In the CHE group, patients (*n* = 178, 85 F and 93 M) had a mean age at diagnosis of primary cancer of 6.9 ± 4.7 years (median 7.2 years, IQR 3.8–11.6 years) and they underwent thyroid screening examinations at a mean age of 13.2 ± 5.6 years. At that time, they were “off therapy” from 6.3 ± 5.2 years (median 5.7 years, IQR 2.3–9.1) and were periodically followed for 8.5 ± 6.1 years (median 7.8 years). In the CHE+RTE group, patients (*n* = 165, 82 F and 83 M) had a mean age at diagnosis of primary cancer of 8.9 ± 4.7 years (median 7.2 years, IQR 3.8–11.6) and, at the time of their first thyroid screening, they were off-therapy from 10.4 ± 9 years (median 7.9 years, IQR 2.6–14.1) with a total mean follow-up of 17.6 ± 12.9 years (median 10 years, IQR 5.2–17).

A functional and/or structural thyroid disease was diagnosed after a mean time of 12.8 ± 11.8 years (median 11.0 years) in 146 of the 343 patients (42.5%), and in a higher percentage, statistically significant, in the CHE+RTE group (103/165 patients, 62.4%, 56 F and 47 M) than in the CHE group (43/178 patients, 24.2%, 28 F and 15 M) (*p* = 0.0001) (Table 2). It is note-worthy that 71 of 343 patients (20.7%), with no evidence of thyroid abnormalities at their first evaluation, developed a thyroid disease during the follow-up (23 in the CHE group and 48 in the CHE+RTE group).

The impact of gender was evaluated in both groups, showing a significantly higher incidence of thyroid cancer in women only in the CHE+RTE group (Appendix A).

Functional Thyroid Diseases (Table 3).

Hypothyroidism: In our series, 54 of 343 patients (15.7%, 30 F and 24 M) were affected by primitive hypothyroidism, 20/178 (11.2%, 13 F and 7 M) in the CHE group vs. 34/165 (20.6%, 17 F and 17 M) in the CHE+RTE group (*p* = 0.01), with a mean age at diagnosis of 16.4 years and after a mean interval of 7.5 years from treatment. In 35 of 54 patients (64.8%), subclinical hypothyroidism was detected (normal thyroid hormone levels with serum TSH > 5 mcU/mL).

In the CHE+RTE group, the mean radiation dose in patients who developed hypothyroidism was 40.5 Gy, administered at a mean age of 10.9 years. The mean latency of hypothyroidism onset after treatment was not statistically different in both groups (7.8 ± 6.8 years in the CHE vs. 7.1 ± 6.1 years in the CHE+RTE group, *p* = 0.4). 

Only in 9 of 54 patients (16.7%) was hypothyroidism associated with elevated serum levels of AAT and AbTPO.

Hyperthyroidism: Graves’ disease was diagnosed in 4 of 343 patients (1.17%, 3 F and 1 M), all cases with clinical hyperthyroidism.

Two of them (1 F and 1 M) were treated with CHE+RTE (RTE dose of 20 and 25 Gy), and Graves’ disease was diagnosed after 4.2 and 5.2 years from primary cancer. The other two female patients (1 for each group) developed hyperthyroidism and concomitant thyroid carcinoma. They were treated for a primary tumour at 14 years and 19 years, and thyroid cancer developed after 11.3 years and 7.9 years, respectively, later. TRAb were positive in both patients, and in one case, orbitopathy was also present.

Autoimmune Thyroiditis: Of 343 patients, 8 (2.3%, 4 F and 4 M) developed chronic thyroiditis with normal thyroid function, 4 (50%, 2 F and 2 M) in the CHE and 4 (50%, 2 F and 2 M) in the CHE+RTE group (*p* = 0.9). It was diagnosed at a mean age of 16.9 years, in the average of 9.7 years after primary cancer diagnosis and treatment. 

Morphological Thyroid Alterations.

Ultrasound Pattern: these data are shown in Table 4.

Thyroid Nodules: Of 343 patients, 63 (18.4%, 31 F and 32 M) developed thyroid nodules (11 with concomitant hypothyroidism).

Thyroid nodules were found in a higher percentage, statistically significant, in the CHE+RTE group (49 patients, 29.7%, 21 F and 28 M) than in the CHE group (14 patients, 7.9%, 10 F and 4 M) (*p* = 0.0001). In the CHE group, the mean latency time for the onset of nodules was not different in the two groups (CHE group: 12.4 ± 5.9, and CHE+RTE group: 14.1 ± 9.8 years from the primary cancer treatment) (*p* = 0.6). In these patients, the mean radiation dose was 28.9 Gy (12–72 Gy). Regarding the patients’ ages at exposure to RTE, 11 patients were less than five years old, 18 patients were between five and ten years, and 20 patients were older than ten years.

FNAB was carried out in 21 of 63 patients (33.3%) with nodules; according to the Italian consensus for thyroid cytology, fourteen cases were classified as “TIR 2”, four cases were “TIR 3A” and three cases were “TIR 3B”. In 42 of 63 patients, FNAB was not performed (in 32 because nodules were smaller than 1 cm; in 10, because all nodules were without suspicious features, and due to the patients’ decision).

Thyroid Carcinoma (Table 5): Of the 343 patients, 30 (8.7%, 23 F and 7 M) developed a differentiated thyroid carcinoma (DTC) (in two cases with concomitant Graves’ disease), with a higher percentage, statistically significant, in the CHE+RTE group 21 (12.7%, 17 F and 4 M) than in the CHE group 9 (5.1%, 6 F and 3 M) (*p* = 0.01). The latency time from primary tumour treatment to diagnosis of thyroid cancer was significantly lower in the CHE group (14.1 ± 8.8 years) than in CHE+RTE group (21.1 ± 11 years) (*p* < 0.01). The mean radiation dose was 27 Gy, given at a mean age of 9.4 years.

DTCs have been diagnosed after a mean follow-up of 19.1 years. In almost all cases (29/30) hystotype was papillary and only in one case follicular.

TNM staging showed a significantly higher percentage of pT1 cancer in the CHE group than in the CHE+RTE group (*p* = 0.04).

Nodal metastases were found in 16 of 30 patients (53.3%) with no statistical difference in the two groups (7/9 vs. 9/21, *p* = 0.11).

^131^I treatment was administered in 23 of 30 patients (76.7%, 17/21 in the CHE+RTE and 6/9 in the CHE group, *p* = 0.6), with a mean administered dose of 96.5 mCi (Table 6).

At the last control visit, 6 of 30 patients (20.0%), 5 in the CHE+RTE and 1 in the CHE group, had persistent disease (lung metastases in three patients and biochemical disease with detectable serum Tg, in the other ones).

## 4. Discussion

In our series of 343 patients treated with CHE or CHE+RTE for paediatric/adolescent tumours, hypothyroidism was the most common functional thyroid alteration with a prevalence of 16% and a period of onset of about seven years after treatment. There was no significant difference between the two groups of treatment. Several previous studies indicate that the most important risk factor for subsequent hypothyroidism is a high radiation dose [11,18]; an high frequency of hypothyroidism was observed in patients exposed to therapeutic doses of 30–70 Gy in the cervical region, and the frequency increases linearly with increasing dose because of progressive thyroid necrosis. In our data, patients who developed hypothyroidism received a higher radiation dose than patients who developed thyroid nodules (40.5 Gy vs. 28.9 Gy).

DNA is one of the most susceptible molecules whose alterations are caused by low radiation doses and consequent cell toxicity, apoptosis and loss of cell cycle control with uncontrolled proliferation [19,20].

It is still not clear what other pathogenic mechanisms cause radiation-induced thyroid dysfunction. Some authors suppose that not only genetic damage but also injury to the small thyroid vessels with relative ischemia of the thyroid gland contributes to thyroid dysfunction. Thyroid cell injury may also be a result of immune-mediated damage.

Radiation-dependent thyroid dysfunctions are more common in subjects with mantle and neck irradiation. However, treatment fields far from the thyroid can also cause thyroid dysfunction because of the limited body surface area of a child [21,22].

Nowadays, new RTE techniques such as stereotactic radiosurgery, three-dimensional conformational irradiation, intensity-modulated RTE and proton therapy allow better and more focused dose distribution with lower doses to the non-target organs [23].

The mean period for hypothyroidism onset observed in our study is consistent with data from previous literature. Hypothyroidism typically occurs with an incidence of 20–30% in the first five years after RTE, with a peak around two to three years after treatment [21]. Some authors observed an earlier onset of hypothyroidism or thyroiditis within three months from the end of RTE, particularly after using high doses. Thyroid damage can occur as an acute/subacute reaction observed within the first three months of RTE, but also as a later reaction, after decades. Hypothyroidism can also occur up to 20 years after RTE, suggesting a longer follow-up in these patients.

In subjects who underwent RTE and had normal antibody levels, a widespread hypo-echogenicity typical of autoimmune thyroiditis was found and this characteristic might be predictive of the development of hypothyroidism [24,25,26,27,28]. However, in other studies, no correlation was found between ultrasound morphological alterations and thyroid function [29], emphasizing that ultrasound and thyroid function tests are complementary.

In our series, hyperthyroidism occurred less frequently with a prevalence of 1% in the CHE+RTE group, a value previously reported in the literature [21]. It usually occurs within three years in subjects exposed to RTE doses greater than 30 Gy, with a prevalence between 0.1% and 2% [30] but has also been observed even 15 years after irradiation when radiant doses ≥20 Gy to the pituitary gland and ≥15 Gy to the neck were administered [21,24].

Hyperthyroidism may be a clinical expression of Graves’ disease or acute thyroiditis. In our series, both hyperthyroid patients developed Graves’ disease approximately five years post-irradiation (radiation doses of 25 Gy and 20 Gy).

Thyroid nodules were a frequent finding in our series. There was a total of 18.4% of cases (63 patients), with a prevalence of 29.7% in subjects undergoing CHE+RTE vs. 7.9% in subjects treated only with CHE (*p* = 0.0001). In other series, the prevalence of benign nodules was extremely variable, and depended on the length of the follow-up as well as the screening method used (neck physical examination or ultrasound) [31,32,33,34,35].

The mean radiation dose administered in our patients who developed benign thyroid nodules was 28.9 Gy (range 12–54 Gy).

It is well known that the patient’s age at the time of irradiation is an important factor for the onset of thyroid nodules; the risk is reduced with an increased age at the time of treatment because the high susceptibility to radiation of the thyroid tissue of young chil-dren progressively decreases with increasing age and is low in adults. This is confirmed in our series; patients who developed thyroid nodules and carcinoma were treated at a younger age than patients who developed hypothyroidism. We observed 30 new cases of thyroid cancer, 9 (5.1%) in the CHE group and 21 (12.7%) in the CHE+RTE group (*p* = 0.01), confirming that thyroid nodular pathology, both benign and malignant, is more frequent after CHE+RTE treatment than after only CHE treatment.

Our data also show a greater predisposition of female irradiated subjects to the development of thyroid cancer. This finding needs to be confirmed by other studies.

The risk of thyroid cancer in children treated with radiation therapy is 15 to 30 times that of subjects not exposed to radiation, and is five-fold greater when children are treated before 4 years old versus 10–14 years old [36]. The risk rises ten-fold for those treated before one year of age. The effect of a radiation dose above 0.05–0.1 Gy increases the risk of thyroid cancer linearly up to 30 Gy and then decreases again at higher doses because of the cell killing possibility with a prevalence of hypothyroidism rather than mutagenicity [18,37,38].

The correlation between irradiation and thyroid carcinoma is analyzed in the study known as “analysis of the 12 studies”, which evaluated 3.4 million children and adolescents exposed to ionizing radiation and in which 927 thyroid carcinomas were identified [36]. In that study, the risk of developing a thyroid carcinoma increased after exposure to a dose of at least 2–4 Gy, remained stable between 10 and 30 Gy and declined, although still remaining significantly high, at a higher dose of 50 Gy. Interestingly, the risk remained elevated for a long period after irradiation up to 50 years [20,39]. Another analysis also showed that a risk persisted for 45 years after exposure for very low doses under 0.2 and 0.1 Gy [40].

However, there is no evidence of a dose for which the risk is insignificant, and the risk was also present for doses under 0.1 Gy. In our study, patients who developed a thyroid carcinoma underwent a mean radiation dose of 28.9 Gy, but this included a wide range (from 15–54 Gy). 

Thyroid carcinomas generally appear at least 5–10 years after exposure to radiation, and this is in accordance with our series where thyroid carcinoma had a latency time of 3.9–42 years (mean 19.1 years). As reported in previous studies, latency time is also related to patient age at the time of irradiation [36].

It is still debated whether radiation-induced thyroid carcinomas have similar histopathological characteristics and similar clinical evolution to sporadic thyroid carcinomas occurring in the paediatric/adolescent stage.

Literature data report that radiation-induced thyroid carcinomas have a favourable prognosis despite the relatively high frequency of lymph node metastases and a clinical evolution similar to sporadic thyroid cancer [41].

This outcome may contribute to the early diagnosis that can be obtained with the periodical screening programs in cancer survivors. In most cases, radiation-induced carcinomas are well-differentiated carcinomas with a papillary histotype, and this also occurred in our series. Younger children usually have a solid or follicular subtype of PTC with an aggressive behaviour in addition to a shorter latency period, whereas adolescents have a more frequently classical PTC at low risk of progression in addition to a longer latency period.

After analysing 1500 cases of thyroid carcinomas in children under 15 years of age that were diagnosed between 1990–2000 after the Chernobyl accident, it was observed that in 94% of cases, the histotype was papillary and the risk of cancer decreased with increasing age at radiation exposure. Furthermore, papillary carcinomas had specific characteristics such as bilaterality, multifocality and extrathyroidal extension [42]. The aggressiveness of the disease was also closely related to the patient’s age at the time of irradiation; in fact, a younger age at the time of exposure had an inversely proportional correlation with the extra-thyroidal extension of tumour, broader lymph node involvement, and a greater probability of distant metastasis [43]. These findings were not confirmed in our patients.

The role of genetic predisposition should also be consistent. For instance, alterations of the DNA repair mechanisms can confer greater susceptibility of the thyroid cell genome to the destabilizing effect of the radiation, favouring the accumulation of mutations and the subsequent neoplastic transformation.

A peculiar characteristic of radiation-induced carcinomas is the presence of mutations of the proto-oncogene RET (RET/PTC rearrangement).

A prevalence of 50–80% of PTC occurred after the Chernobyl accident; however, these data are not confirmed in other studies [44]. Among the three types of mutations of the oncogene in radiation-induced carcinomas, RET/PTC-3 mutations prevail. 

It is still a matter of discussion whether the detrimental role of CHE on the thyroid is only additional to that of RTE, if it significantly alters thyroid biology per se [12,22,45] or if it is without significant effects [14,15,46].

The great variety of drugs used to treat paediatric cancers and the different protocols in terms of time and dose could explain the variability of observed consequences. The thyroid gland is less susceptible to the detrimental effect of CHE than other endocrine glands such as the gonads. In a previous study carried out in 177 children with Hodgkin’s disease, those treated exclusively with CHE did not develop any thyroid dysfunction, while patients treated only with RTE or with CHE+RTE developed hypothyroidism [47]. Other studies showed that the incidence of hypothyroidism and thyroid cancer increased in patients treated with alkylating agents (busulfan, cyclophosphamide) due to cell damage and mutagenesis that affected the DNA [11,13,48].

## 5. Conclusions

In conclusion, our study indicates that:There was a greater prevalence of functional (hypothyroidism) and morphological (nodules and thyroid carcinomas) thyroid alterations observed in the CHE+RTE group compared to the CHE group.The mean radiation dose administered at the time of childhood malignancy was significantly higher in patients who developed hypothyroidism.A young age at the time of irradiation increased the risk of secondary thyroid cancer.The latency period between childhood cancer treatment and the onset of thyroid alterations was longer for the appearance of thyroid nodules than for the occurrence of hypothyroidism.The prevalence of morpho-functional thyroid alterations after CHE was higher than that reported in the literature in the general paediatric and adolescent population. Further research is needed to define the effects of some chemotherapeutic agents on the onset of thyroid diseases.

Our data suggest the role and the clinical benefit 
of morpho-functional thyroid screening in paediatric/adolescent can-cer 
survivors and that careful lifetime follow-ups are mandatory.

## Figures and Tables

**Table 1 children-08-00767-t001:** Primary paediatric/adolescent cancer.

Primary Cancer	(*n* = 343)	(%)
Acute Lymphatic Leukaemia	186	54.2
Hodgkin’s Disease	63	18.4
Non Hodgkin’s Disease	28	8.2
Acute Myeloid Leukaemia	11	3.2
Medulloblastoma	7	2.0
Rhinopharyngeal Carcinoma	5	1.5
Ewing Sarcoma	6	1.8
Wilms’s Tumour	5	1.5
Others tumors	32	9.2

**Table 2 children-08-00767-t002:** Incidence of Thyroid Disease in 343 patients treated for paediatric/adolescent cancer.

	All Cohort (*n* = 343)	CHE Group (*n* = 178)	CHE+RTE Group (*n* = 165)	*p*
Thyroid Disease	146 (42.5%)	43 (24.2%)	103 (62.4%)	0.0001
-Hypothyroidism *	54 (15.7%)	20 (11.2%)	34 (20.6%)	0.001
-Hyperthyroidism **	4 (1.17%)	1 (25.0%)	3 (75.0%)	ns
-Chronic Thyroiditis	8 (2.3%)	4 (50%)	4 (50%)	ns
-Thyroid nodules *	63 (18.4%)	14 (7.9%)	49 (29.7%)	0.0001
-Thyroid cancer **	30 (8.7%)	9 (5.1%)	21 (12.7%)	0.01

* 1 patients with Hypothyroidism also presented thyroid nodules (4 in CHE and 7 in CHE+RTE group). ** 2 patients with presented thyroid cancer (1 in CHE and 1 in CHE+RTE group).

**Table 3 children-08-00767-t003:** Incidence of functional thyroid disease in patients treated for paediatric/adolescent cancer.

	CHE Group (*n* = 178)	CHE+RTE Group (*n* = 165)	*p*
Hypothyroidism			
-N. of cases	20 (11.2%)	34 (20.6%)	0.01
-F/M ratio	1.8/1.0	1.0/1.0	
-Age at cancer *	6.2 (±4.8)	10.2 (±3.5)	0.13
-Latency period *	7.8 (±6.8)	7.1 (±6.1)	0.4
-Follow-up *	10.2 (±5.7)	13.1 (±9.1)	0.12
-RTE Dose	-	40.5 Gy	
Hyperthyroidism			
-N. of cases	1 (0.6%)	3 (1.8%)	Not performed
-F/M ratio	1.0/0	2/1.0.0	
-Age at cancer *	14.0	14.7 (±4.6)	
-Latency period *	11.3	4.7 (±0.7)	
-Follow-up *	12.0	8.6 (±5.4)	
-RTE Dose	-	25.0 Gy	
Chronic Thyroiditis			
-N. of cases	4 (2.2%)	4 (2.4%)	0.9
-F/M ratio	1.0/1.0	1.0/1.0	-
-Age at cancer *	5.8 (±3.3)	8.5 (±3.1)	0.07
-Latency period *	9.3 (±9.2)	10.1 (±7.1)	0.8
-Follow-up *	11.04 (±8.2)	13.9 (±6.1)	0.8
-RTE Dose	-	24.5 Gy	-

* Years Mean (±SD).

**Table 4 children-08-00767-t004:** Ultrasound pattern according to thyroid status.

	Patients (%)	CHE *n*. 186 (%)	CHE+RTE *n*.157 (%)
Pattern 1	140 (40.8)	91 (48.9)	49 (31.2)
Pattern 2	184 (53.6)	90 (48.4)	94 (59.9)
Pattern 3	19 (5.6)	5 (2.7)	14 (8.9)
Normal thyroid function	197 (57.4)	135 (72.6)	62 (39.5)
-Pattern 1	92	68	24
-Pattern 2	102	65	37
-Pattern 3	3	2	1
Hypothyroidism	43 (12.5)	16 (8.6)	27 (17.2)
-Pattern 1	2	1	1
-Pattern 2	30	14	17
-Pattern 3	11	2	9
Chronic Thyroiditis	8 (2.3)	4 (2.2)	4 (2.5)
-Pattern 1	None	None	None
-Pattern 2	5	3	2
-Pattern 3	3	1	2
Hyperthyroidism	2 (0.6)	none	2 (1.3)
-Pattern 1	1		1
-Pattern 2	1		1
-Pattern 3	none		none
Thyroid nodules	52 (15.2)	10 (5.4)	42 (26.8)
-Pattern 1	16	5	11
-Pattern 2	36	5	31
-Pattern 3	none		none
Thyroid cancer	30 (8.7)	9 (4.8)	21 (13.4)
-Pattern 1	28	8	20
-Pattern 2	2	1	1
-Pattern 3	none	none	none

**Table 5 children-08-00767-t005:** Incidence of Morphological thyroid disease in patients treated for paediatric/adolescent cancer.

	CHE Group	CHE+RTE Group	*p*
	(*n* = 178)	(*n* = 165)	
Thyroid Nodules			
- N. of cases, (%)	14 (7.9%)	49 (29.7%)	<0.01
- F/M ratio	2.5/1.0	0.8/1.0	
- Age at cancer *	6.8 (±5.0)	8.4 (±4.6)	0.3
- Latency period *	12.4 (±5.9)	14.1 (± 9.8)	0.6
- Follow-up *	16.7 (±8.4)	21.8 (±12.9)	0.2
- RTE Dose	-	28.9 Gy	
Thyroid Cancer			
- N. of cases, (%)	9 (5.1%)	21 (12.7%)	0.01
- F/M ratio	2.0/1.0	4.2/1.0	
- Age at first cancer *	10.8 (±5.2)	8.6 (±5.9)	0.3
- Latency period *	14.1 (±8.8)	21.1 (±11)	0.01
- Follow-up *	16.5 (±7.5)	29.0 (±13)	0.02
- RTE Dose	-	27 Gy	
Thyroid cancer and Hyperthyroidism			
- N. of cases, (%)	1 (0.6%)	1 (0.6%)	
- Age at first cancer *	14	19	
- Latency period *	11.3	7.9	
- Follow-up *	12	27	
- RTE Dose		20 Gy	

* years mean (±SD).

**Table 6 children-08-00767-t006:** Characteristics of 30 DTC patients.

	All Cohort	CHE Group (*n* = 178)	CHE+RTE Group (*n* = 165)	*p*
- N. of cases	30	9 (5.1%)	21 (12.7%)	0.01
-Mean age (±SD)	28.4 (±9.1)	25 (±8.2)	29.8 (±11.4)	0.07
- F/M ratio	3.3/1.0	2.0/1.0	4.2/1.0	-
- Mean size cm (±SD)	1.5 (±1.4)	0.9 (±0.2)	1.8 (±1.5)	0.2
- T Status (*n*., %)				
Tx	2 (6.7%)	0	2 (9.5%)	-
T1	18 (60.0%)	8 (88.9%)	10 (47.6%)	0.04
T2	4 (13.3%)	0	4 (19.0%)	−
T3	6 (20.0%)	1 (11.1%)	5 (23.7%)	0.6
-N Status (*n*., %)				
N0/Nx	14 (46.7%)	2 (44.4%)	12 (57.1%)	0.11
N1a	9 (30.0%)	4 (22.2%)	5 (23.8%)	0.38
N1b	7 (23.3%)	3 (33.4%)	4 (19.1%)	0.64
-M Status (*n*., %)				
M0/Mx	28 (93.3%)	9 (100%)	19 (90.4%)	1
M1	2 (6.7%)	0	2 (9.6%)	1
-Stage (*n*., %)			17 (80.9%)	
I	26 (86.7%)	9 (100%)	3 (14.2%)	1
II	3 (10.0%)	0	1 (4.9%)	0.54
III	1 (3.3%)	0		1

## Data Availability

Data are available in the Garibaldi-Nesima Medical Center archive.

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
