# Peer review of "Risk of Benign and Malignant Thyroid Disorders in Subjects Treated for Paediatric/Adolescent Neoplasia: Role of Morphological and Functional Screening"

_children, 2021, doi:10.3390/children8090767_

Round 1

Reviewer 1 Report

Very interesting and complete manuscript. The manuscript provides an important analysis of the occurrence of thyroid benign and aggressive tumours, an issue that was not taken into account for many years. The importance of the manuscript is the patients follow up and the consecutive analysis. of the type of tumours, especially in those patients treated with radiotherapy and chemotherapy independently of the original tumour. The report itself is clearly appreciated by the paediatricians and pediatric oncologists although part of this cohort surely will be studied for other details. Thus, I think alerting the clinicians which may benefit patients that were treated in similar schemes and they may be at risk of developing thyroid disease.    
The only issue that was not clear is the statistical impact of gender and/or other metabolic changes that may occur in both groups, changes in BMI, sex hormones, etc. However, those can be a part of another manuscript .

Author Response

Thank you for your comments and suggestions.

We evaluated, as suggested, the impact of gender in both groups and found a significant higher incidence of thyroid cancer in women only in CHE+RTE group (Supplementary Table 1). In all other cases we did not find any significant differences.

Reviewer 2 Report

I read with interest Giulia Sapuppo et al. article entitled Risk of Benign and Malignant Thyroid Disorders in Subjects Treated for Paediatric/Adolescent Neoplasia: Role of Morphological and Functional Screening

The authors, present their retrospective study on the the incidence of thyroid alterations evaluated in a series of 343 patients treated with Chemo (CHE) and Radiotherapy (RTE).

Up to their results patients treated with CHE + RTE the prevalence of hypothyroidism and nodular pathology are significantly greater than in patients treated with CHE.

Since the largely known radiation effect on thyroid gland the results presented are not surprising and unfortunately the scientific soundness is low.

The article is quite well written but is plenty of small formal/editing errors (“ - “ appears frequently in the text or lack of space between words) and notes written (also in italian) left during the revision of the paper p.ex cfr line 239. The article doesn’t seem a definitive version of the paper and seems unlikely that 12 (twelve authors?) have red the final submitted version of the paper.

In conclusion the main concern in on the scientific soundness that is low moreover I think that the paper should be deeply re-worker and revised.

As a suggestion reconsider the number of authors: twelve authors with 4 first co-authors seems frankly too much for this kind of work and affects the credibility of the paper.

In details I have few comments to make to the authors:

Comments:

Lines 48,105,286: provide bibliography

Line 101 define better the ultrasound evaluation and patterns as in literature (eu-tirads?)

Line 239 how was decided to perform or not perform the FNA?

Line 251-254 are comments? Revisions suggestions? This is not pertinent in the article

Table 5 and 6 should be reformatted they appear not clear  

Line 309 this sentence is not clear at all (compared with literature or CHE group?)

Best regards

Author Response

I read with interest Giulia Sapuppo et al. article entitled Risk of Benign and Malignant Thyroid Disorders in Subjects Treated for Paediatric/Adolescent Neoplasia: Role of Morphological and Functional Screening

The authors, present their retrospective study on the the incidence of thyroid alterations evaluated in a series of 343 patients treated with Chemo (CHE) and Radiotherapy (RTE).

Up to their results patients treated with CHE + RTE the prevalence of hypothyroidism and nodular pathology are significantly greater than in patients treated with CHE.

Since the largely known radiation effect on thyroid gland the results presented are not surprising and unfortunately the scientific soundness is low.

The article is quite well written but is plenty of small formal/editing errors (“ - “ appears frequently in the text or lack of space between words) and notes written (also in italian) left during the revision of the paper p.ex cfr line 239. The article doesn’t seem a definitive version of the paper and seems unlikely that 12 (twelve authors?) have red the final submitted version of the paper.

In conclusion the main concern in on the scientific soundness that is low moreover I think that the paper should be deeply re-worker and revised.

As a suggestion reconsider the number of authors: twelve authors with 4 first co-authors seems frankly too much for this kind of work and affects the credibility of the paper.

We apologize for the typos (in copying within the spaces of the template) and language mistakes; the entire manuscript has been revised.

In details I have few comments to make to the authors:

Comments:

  • Lines 48,105,286: provide bibliography

As suggested we provided bibliography and revised the entire manuscript.

  • Line 101 define better the ultrasound evaluation and patterns as in literature (eu-tirads?)

Thyroid ultrasound examination included the evaluation of ultrasound patterns of the whole thyroid gland, subdivided into grade 1 (normal echogenicity, 2 moderately hypoechoic or 3 diffusely hypoechoic. Nodules were evaluated by eu-tirads classification.

The sentence has been rephrased as suggested.

  • Line 239 how was decided to perform or not perform the FNA?

FNAb was carried out in 21 of 63 patients (33.3%) with nodules.; 42 of 63 FNAb was not performed (32 as smaller than 1 cm and in 10, all nodules without suspicious features, for patients’ decision).

A sentence has been added at line 108- 109 and line 254.

  • Line 251-254 are comments? Revisions suggestions? This is not pertinent in the article

We apologize for the suggestions by an author not removed

  • Table 5 and 6 should be reformatted they appear not clear  

Tables 5 and 6 have been reformatted

  • Line 309 this sentence is not clear at all (compared with literature or CHE group?)

The sentence has been rephrased as suggested.

Round 2

Reviewer 2 Report

no more comments